# Vpr Is a VIP: HIV Vpr and Infected Macrophages Promote Viral Pathogenesis

**DOI:** 10.3390/v12080809

**Published:** 2020-07-27

**Authors:** Jay Lubow, Kathleen L. Collins

**Affiliations:** 1Department of Microbiology & Immunology, University of Michigan, Ann Arbor, MI 48109, USA; jaylubow@umich.edu; 2Department of Internal Medicine, University of Michigan, Ann Arbor, MI 48109, USA

**Keywords:** HIV, Vpr, Env, macrophages, restriction factor, mannose receptor

## Abstract

HIV infects several cell types in the body, including CD4^+^ T cells and macrophages. Here we review the role of macrophages in HIV infection and describe complex interactions between viral proteins and host defenses in these cells. Macrophages exist in many forms throughout the body, where they play numerous roles in healthy and diseased states. They express pattern-recognition receptors (PRRs) that bind viral, bacterial, fungal, and parasitic pathogens, making them both a key player in innate immunity and a potential target of infection by pathogens, including HIV. Among these PRRs is mannose receptor, a macrophage-specific protein that binds oligosaccharides, restricts HIV replication, and is downregulated by the HIV accessory protein Vpr. Vpr significantly enhances infection in vivo, but the mechanism by which this occurs is controversial. It is well established that Vpr alters the expression of numerous host proteins by using its co-factor DCAF1, a component of the DCAF1–DDB1–CUL4 ubiquitin ligase complex. The host proteins targeted by Vpr and their role in viral replication are described in detail. We also discuss the structure and function of the viral protein Env, which is stabilized by Vpr in macrophages. Overall, this literature review provides an updated understanding of the contributions of macrophages and Vpr to HIV pathogenesis.

## 1. Fundamentals of Human Immunodeficiency Virus

Human immunodeficiency virus type I (referred to as simply HIV throughout this review) is a lentivirus of the Retroviridea family [1]. The viral particle contains two copies of a positive strand RNA genome, a capsid composed of the viral protein Gag, and a lipid membrane derived from the producer cell. The viral protein Env is anchored in the membrane and is responsible for binding the main HIV receptor CD4 and a co-receptor (Figure 1). The complex structure of Env is critical to HIV replication and is described in greater detail below. HIV can only infect cells that express CD4 and a chemokine co-receptor (most often CCR5 or CXCR4), which, to the best of our knowledge, limits viral tropism to immune cells, namely CD4^+^ T cells [2], macrophages [3], dendritic cells (DCs) [4], osteoclasts [5,6], and hematopoietic stem and progenitor cells [7,8].

After entry to the cell, the capsid and viral RNA genome are imported into the nucleus, where the genome is reverse transcribed to produce a double-stranded cDNA copy and the capsid disassembles [9]. Whereas most retroviruses wait for the nuclear membrane to be dissolved during mitosis, the nuclear import capability of HIV enables it to infect non-dividing cells and categorizes it specifically as a lentivirus [10]. The integrated viral genome can, at this point, continue the replication cycle by making new copies of viral proteins and RNA genomes, or it can enter a latent state, in which transcription of the viral genome is suppressed. The latent state can persist for years or even decades before being reactivated, at which point the genome can produce fully functional virions.

When the HIV genome is active, host transcriptional and translational machineries are induced to generate numerous viral proteins and genomes. Host RNA polymerase II transcribes full-length RNA copies of the viral genome, most of which are spliced by host machinery to generate mRNAs specific for a single viral protein. In later stages of infection, full-length viral genomes are exported and serve multiple critical functions. The full-length viral RNA can be translated to produce Gag or Gag–Pol, two polyproteins that are cleaved to form most of the virus’s structural proteins, including the capsid, reverse transcriptase, and integrase. The full-length genome also contains a packaging sequence that causes it to bind the capsid and be packaged into new virions.

In addition to structural and regulatory proteins that are essential to the replication cycle, HIV also encodes accessory proteins that do not directly participate in replication processes, but which alter the host cell environment to maximize virion production and persistence. Vif, Vpu, and Nef counteract numerous innate and adaptive defenses. They enhance infection in cell lines, primary CD4+ T cells, and primary macrophages [11,12,13]. By contrast, the viral accessory protein Vpr primarily enhances infection of macrophages [11,14], which is discussed in greater detail below.

## 2. HIV Infection of Macrophages

### 2.1. Distribution and Variety of Macrophages

There are numerous types of macrophages in the human body that are specialized for their particular role and anatomical location. Some macrophages are derived from monocytes, which circulate in the blood and can differentiate into several different cell types, including macrophages and dendritic cells [15]. Monocytes, in turn, are derived from hematopoietic progenitor cells in the bone marrow [16]. A separate population of macrophages, collectively referred to as tissue resident macrophages, are derived from myeloid progenitors that migrate from the embryonic yolk sac within the first weeks of embryogenesis [17]. These precede the development of monocytes derived from bone marrow and persist in tissues for the life of an organism [18]. The fraction of macrophages that are monocyte-derived and the fraction that are of embryonic origin vary across anatomical sites and are a matter of ongoing study. Because the morphology and protein expression of different types of macrophages can vary significantly, they are often identified by the surface markers CD68 [19], CD11b [20], and CD14 [21,22], although immature monocytes also express CD14 [23].

### 2.2. Macrophage Functions

A wide array of receptors on the surface of macrophages allow macrophages to detect and respond to various stimuli (reviewed in Reference [20]). A major target of macrophages are host cells displaying apoptotic markers, which are recognized by numerous receptors, phagocytosed, and degraded [24]. This allows the clearance of dead and dying host cells and avoids activating an inflammatory response [25].

Macrophages are among the first lines of defense in the innate immune system, primarily due to their ability to phagocytose pathogens. Phagocytosis can be mediated by a wide variety of receptors on the plasma membrane, including pattern-recognition receptors (PRRs), which directly bind pathogen-associated molecular patterns (PAMPs); complement receptors, which bind proteins of the complement system; and Fc receptors, which bind the constant regions of antibodies. Within these categories are numerous subtypes that collectively provide the capacity to phagocytose nearly any class of pathogen, including fungi [26], parasites [27,28], bacteria such as *Staphylococcus aureus* [29] and *Listeria monocytogenes* [30], and viruses such as influenza A virus [31] and foot-and-mouth disease virus [32], among others. For those pathogens that do not express sufficient PAMPs to be phagocytosed directly, soluble factors, including IgG [33], complement proteins, and antimicrobial peptides [34], bind to pathogens and significantly enhance phagocytosis.

Phagocytosis is the first step in the macrophage’s role as a professional antigen-presenting cell (APC). Macrophages, along with dendritic cells (DCs) and B cells, degrade ingested proteins and present the resulting peptides on MHC class I and class II molecules to activate T-cell responses. This process has been demonstrated in numerous types of macrophages, including Kupffer cells [35], adipose tissue macrophages [36], and alveolar macrophages [37]. Presentation of exogenous antigens on MHC class I, a process called cross-presentation, only occurs in professional APCs, and this process in macrophages and DCs is crucial for activating naïve CD8^+^ cytotoxic T lymphocytes [38,39,40].

Macrophages also regulate inflammation by secreting various cytokines in response to different stimuli. In vitro, classically activated macrophages, also known as M1, secrete numerous pro-inflammatory cytokines, most notably IL-12 [41], which supports Type 1 T helper cells [42,43]. Anti-inflammatory macrophages, also known as M2, secrete IL-10 [41], which decreases expression of pro-inflammatory cytokines by macrophages [44] and suppresses activation of T cells [45]. These two activities are thought to be crucial to initiating and later resolving inflammation following infection.

### 2.3. Macrophage Models

Due to their significance in homeostasis, immunity, and infection by various pathogens, models that facilitate investigation of macrophages are in high demand. Explants of human tissue have been used to study macrophages in the lungs [46], adipose tissue [47], tonsils [48], urethra [49], foreskin, and cervix [50]. These models contain mature macrophages in their natural microenvironments, but their use is limited by cost and difficulty of obtaining human tissues. For models of isolated macrophages, alveolar macrophages obtained from bronchoalveolar lavage can also be used for ex vivo studies [51,52]. Primary macrophages can also be generated from monocytes, of which tens of millions can be obtained from peripheral blood of an individual. Monocyte-derived macrophages are stimulated in vitro by using granulocyte-macrophage colony stimulating factor (GM-CSF), often in conjunction with other cytokines, such as macrophage colony stimulating factor (M-CSF). Different stimulation conditions cause differential expression of key macrophage proteins, including PRRs, CD4, and chemokines receptors [53]. Additionally, there is significant variability in macrophages obtained from different donors [54]. Differences between stimulation protocols and among donors should be considered when evaluating or replicating experiments using monocyte-derived macrophages.

A handful of myeloid cell line models exist and provide useful clues about macrophage biology. The most common is the THP-1 line, derived from a patient with monocytic leukemia [55]. These monocyte-like cells can be induced to differentiate into macrophage-like cells with various combinations of phorbol-12-myristate-13-acetate (PMA), lipopolysaccharide (LPS), and vitamin D3. These stimulated THP-1s replicate some but not all features of primary monocyte-derived macrophages [56].

### 2.4. Evidence for HIV Infection of Macrophages

In addition to their role in clearing pathogens, macrophages can also be the target of infection. Due to their natural ability to phagocytose foreign bodies, they are particularly vulnerable to intracellular pathogens, including *Mycobacterium tuberculosis* [57], *Listeria monocytogenes* [30], Dengue virus [58], simian immunodeficiency virus (SIV) [59], and HIV [3].

HIV has been detected in macrophages isolated from infected humans at many anatomical sites, including lymph nodes [60], brain [61], urethra [62], and liver [63,64]. In infected persons, viral replication in macrophages is the dominant source of viremia after an infected person’s CD4+ T cell population falls [65], demonstrating that macrophage infection is productive and long lasting. Given that an infected person’s viral load correlates strongly with disease progression [66], virion production by long-lived infected macrophages likely accelerates the development of symptoms. Recent experiments have demonstrated that macrophages can sustain HIV infection independently of CD4^+^ T cells. Infections of humanized mice that produce human macrophages but not human T cells replicate significant aspects of HIV infection in humans [67]. The macrophage-only mouse can sustain infection for at least 10 weeks, at which point HIV was detected in numerous tissues throughout the body [67]. A follow up study using the same mouse model found that antiretroviral treatment led to a rapid decrease in viremia. After treatment interruption, viral rebound was observed in three out of nine mice, indicating that macrophages can act as a long-lived reservoir [68]. An important caveat of this study is that the humanized mouse produces human macrophages derived from monocytes, but its tissue resident macrophages are exclusively of murine origin and therefore cannot be infected by HIV. Because tissue-resident macrophages are long-lived and self-renewing, an in vivo model that lacks these cells may not fully recapitulate the long-term role of macrophages in HIV infection.

### 2.5. HIV Infection of Macrophages Contributes to Pathogenesis

Whereas most HIV-infected T cells undergo apoptosis within days of infection, HIV-infected macrophages survive for weeks or months [69]. This is likely because they are less susceptible to the cytopathic effects of the virus [70,71]. The number of alveolar macrophages in humans is virtually unaffected by HIV infection, but their ability to phagocytose *Pneumocystis carinii* is inhibited [72]. Similar inhibition has been noted in HIV-infected macrophages’ ability to phagocytose *Candida albicans* [73], *Toxoplama gondii* [74], and *Plasmodium falciparum* [28], which likely contributes to increased susceptibility to opportunistic infections. Additionally, HIV infection polarizes macrophages into a pro-inflammatory M1 phenotype, as measured by cytokine secretion and surface-marker expression [75]. Experiments in rhesus macaques demonstrated that SIV infection enhances macrophage-mediated inflammation of the liver [76], which may explain why HIV-infected persons have an increased incidence of liver disease [77]. Inflammation caused by HIV-infected macrophages is also thought to be the primary driver of HIV-associated atherosclerosis [78] and HIV-associated neurocognitive disorders [79]. 

In addition to direct effects on macrophages, HIV infection of macrophages contributes to overall pathogenesis by increasing viral loads in infected individuals. Replication competent virus has been recovered from monocytes and macrophages of people receiving antiretroviral therapy [80], suggesting that they may contribute to viral rebound after therapy is withdrawn. In ex vivo experiments using explants of human lymphoid tissue, infection of macrophages boosts virion production significantly, despite comprising a small fraction of infected cells [81]. This suggests that macrophages amplify infection in other cell types, most notably CD4^+^ T cells.

There is significant evidence that HIV spreads via cell-to-cell contact [82] and that this method is important in macrophages [83]. Direct cell-to-cell spread is mediated through a virological synapse, a structure that is formed when viral Env on the plasma membrane of an HIV-infected cell binds CD4 on a neighboring uninfected target [84]. This connection allows multiple viral particles to be transmitted to a single target cell, greatly enhancing the likelihood that the virus establishes a successful infection [85]. Transmission via synapses is resistant to neutralization by antibodies [86], meaning it may be particularly difficult to inhibit by an antibody-based HIV vaccine. This form of transmission mediates the spread of HIV from macrophages to CD4^+^ T cells [87], which is much more efficient than infection of CD4^+^ T cells by cell free virus [88,89]. In addition, phagocytosis of infected CD4^+^ T cells by macrophages leads to greater infection of macrophages than does incubation with cell-free virus [90]. Direct contact between infected CD4^+^ T cells and uninfected macrophages can also cause cell fusion, which ultimately leads to the formation of giant multinucleated cells [91]. This process does not require reverse transcription, which evades the restriction imposed by SAMHD1 [92,93,94] and thereby increases infection of macrophages, DCs, and osteoclasts [95]. Combined, these findings demonstrate that HIV replication is significantly enhanced by cell-to-cell transmission between macrophages and T cells, which explains the earlier observation that the presence of macrophages boosts viral burden in lymphoid tissues [81]. Cell-to-cell spread from macrophages is restricted by mannose receptor (MR), which binds to mannose residues of HIV Env [89]. This restriction is alleviated by the viral accessory protein Vpr [14,88], a process that is described in greater detail in the Vpr section below.

### 2.6. The Role of Macrophages in HIV Transmission

Whether and to what degree macrophages contribute to HIV transmission is a matter of much debate. Early studies of tropism found that most HIV isolates from early stages of infection did not infect T-cell lines and were thus thought to be “macrophage tropic” [96,97]. More recent studies found that this observation, though correct, was misleading, because, unlike primary CD4^+^ T cells, most T-cell lines do not express CCR5, the co-receptor used by most viral isolates [98]. Viruses isolated within the first few weeks of transmission, called transmitted/founder or T/F viruses, can infect primary CD4^+^ T cells and macrophages [99]. This study found that over 4–14 days T/F viruses replicated to higher titers in T cells than in macrophages, suggesting that T/F viruses have a greater capacity for infection of T cells, which express high levels of CD4 [98]. 

There is significant in vivo evidence that indicates macrophages may facilitate transmission. Macrophages are present in semen, urethra, foreskin, vaginal mucosa, rectal mucosa, and cervical mucosa [100], making them a potential source or target of HIV in a sexual transmission event. In infected male humans, HIV has been found in CD4^+^ T cells and macrophages in semen [101] and in the urethra [62]. In infected macaques, SHIV (an SIV–HIV hybrid) was found in macrophages in the testes and epididymis [102]. In infection simulation experiments using explants of human cervical tissue, HIV was found in macrophages and T cells [103,104]. In both of these studies, infected macrophages were more abundant than infected T cells, although the strain used (BaL) is notably macrophage tropic. One study [103] attempted the same experiment with two lab-adapted T-cell tropic clones (IIIB and RF) and saw very little infection of any cell type. Similar studies using HIV clones isolated early in infection would be highly informative.

## 3. Mannose Receptor

### 3.1. Cell Biology of Mannose Receptor

Mannose receptor (MR) is a multidomain, multifunctional pattern-recognition receptor that is highly expressed on macrophages and a handful of other cell types, including dendritic cells and epithelial cells [105,106]. It is a type I transmembrane protein composed of an N-terminal cysteine rich domain, which binds sulfonated sugars [107]; a fibronectin type II domain, which binds collagen [108,109]; and eight C-type lectin domains, which bind mannose and several other hexoses (Figure 2). When the domains are tested in isolation, lectin domain 4 has by far the highest affinity for mannose, but when multiple domains are tested together, a fragment containing domains 4–8 has higher binding affinity still, indicating that binding is multivalent [110,111]. That multiple MR domains can bind the same ligand simultaneously is also supported by the observation that MR has higher affinity for multiply branched polysaccharides than linear ones [112].

MR expression and function has been studied extensively in macrophages. It is estimated that, at any given moment, approximately 100,000 copies of MR are available for binding on the surface of a macrophage, and that five times that number are in internal compartments [113]. Mannose-containing particles are internalized within five minutes of binding, and this process was not inhibited by cycloheximide, which blocks translation of new proteins, indicating that newly synthesized MR is not required for MR activity [113]. This suggests that mannose receptor recycles to the cell membrane to bind and endocytose cargo repeatedly. 

### 3.2. Microbial Interactions with Mannose Receptor

MR’s role as a pattern-recognition receptor that leads to internalization of pathogens is central to macrophages’ function as immune cells. MR binds polysaccharides and lipopolysaccharides found in capsules and cell walls of numerous species of bacteria, leading to endocytosis [114]. MR also binds fungi such as the pathogenic *Cryptococcus neoformans,* leading to activation of CD4^+^ T cells. This process is crucial for antifungal immunity, as demonstrated by the observation that mice lacking MR *(MRC1-/-)* are significantly less likely to survive *C. neoformans* infection [115]. 

MR can also be hijacked by pathogens to evade immunity and enhance pathogenesis. A well-characterized example is *Mycobacterium tuberculosis,* which produces lipoarabinomannans to bind MR and inhibit production of IL-12, TNF-α, and TGF-β [116,117]. MR mediates macrophage entry for several notable pathogens, including *Streptococcus pneumoniae* [118] and Dengue virus [119]. Treatment of macrophages with vitamin D3 lowers MR expression, which reduces Dengue virus infection and the release of inflammatory cytokines [120].

HIV binding to MR has been demonstrated in many contexts and significantly alters infection and viral replication. One group found that MR binds HIV on the plasma membrane of macrophages and transfers virions to CD4^+^ T cells, a process known as trans-infection [121]. It was recently demonstrated that this interaction also enhances cis-infection of MR-expressing macrophages [89]. This activity is analogous to that of DC-SIGN, which enhances trans-infection by DCs [122] and members of the Siglec family, which enhance both trans- and cis-infection of DCs and macrophages [123,124]. Given its role in enhancing HIV infection, it was somewhat surprising that several groups have observed that infection by HIV or SIV decreases MR expression and MR-mediated phagocytosis [72,75,125,126]. It was recently revealed that this is because, after a productive HIV infection has been established, MR continues to bind Env, which restricts HIV spread [89]. Downmodulation of MR is mediated simultaneously and independently by two HIV accessory proteins: Nef [127] and Vpr [89]. The effects of Vpr on MR and the implications for HIV pathogenesis are discussed in more detail in a separate section below. 

## 4. The HIV Structural Protein Env

The HIV gene *env* encodes a large glycoprotein responsible for mediating binding and entry of the virion to target cells. Entry to a target cell requires the presence of two host proteins, the primary receptor CD4 and one of two potential co-receptors, CCR5 or CXCR4. The receptor CD4 is bound first and induces a conformational change in gp120 that exposes the co-receptor binding site [128]. Binding by CCR5 or CXCR4 induces another conformational change that exposes the fusion peptide, a hydrophobic region of gp41 that inserts into the membrane of the target cell and promotes fusion [129].

### Env Biosynthesis

Env is a large and structurally complex protein which must be produced in the secretory pathway, where it undergoes numerous post-translational modifications (Figure 3). It is a type 1 transmembrane protein, meaning it has an N terminus facing the lumen/extracellular space, a single transmembrane domain, and a C terminus in the cytosol. Env is translated in the lumen of the rough ER, where it is simultaneously glycosylated at numerous asparagine residues [130]. At each of these residues, the enzyme complex oligosaccharyltransferase attaches a glycan tree [131] that is identical for all asparagine-linked (N-linked) glycosylation sites (Figure 4A). In the ER, glucose is removed from the glycan tree, leaving three branches that all terminate in multiple mannose residues (Figure 4B). This form is called oligomannose, Man_5_-_9_GlcNAc_2_, or high-mannose [132].

As most glycoproteins transit through the Golgi, the terminal mannose residues are removed by mannosidases [134]. These are replaced by many possible sugar conformations composed of fucose, galactose, sialic acid, and N-acetylglucosamine [135], which are collectively known as “complex glycans” (Figure 4C). 

Unlike healthy host proteins, HIV Env is not fully processed by this pathway. Due to the unusually high number of glycosylation sites on Env and its homotrimerization, some of the glycan trees cannot be accessed by mannosidases [132]. Studies of Env in which N-linked glycosylation sites are removed one at a time found that the loss of certain sites in and near a dense cluster of N-linked glycosylation sites known as the “mannose patch” leads to >25% loss of oligomannose, presumably by increasing accessibility of neighboring glycans [136]. In the final step of Env synthesis, the full-length polyprotein known as gp160 is cleaved by the host protease furin to produce gp120 and gp41. Cleavage occurs in the trans-Golgi and is required for Env function [137]. 

Due to its distinctly non-eukaryotic glycan structures and position as the only viral protein outside the viral membrane, Env is the target of numerous innate and adaptive immune mechanisms. Env is the most common target of antibodies produced in infected humans [138] and has therefore been the target of most efforts to generate an antibody-based vaccine [139]. Env can also be targeted by innate mechanisms, which was first demonstrated by the observation that interferon-α (IFN α) treatment reduces infectivity of virions by inhibiting Env assembly [140]. Since then, two interferon-inducible innate immune factors have been shown to restrict Env. Guanylate binding protein 5 (GBP5) is expressed in the Golgi, following interferon treatment, and prevents proper Env processing [141]. Interferon-inducible transmembrane protein 3 (IFITM3) binds Env and reduces infectivity of virions [142,143].

## 5. The HIV Accessory Protein Vpr

The HIV gene *vpr* produces a 96 amino acid 14 kDa protein [144] that has many known biochemical functions, but its role in the viral replication cycle has been a subject of much debate. Evolutionary analysis provides evidence that it is crucial for infection in vivo. Vpr is conserved in all primate lentiviruses [145], and individuals infected with Vpr mutants are extremely rare. The first known case was an occupational transmission of HIV HxB2, which contains a truncation in Vpr at amino acid 78. This mutation reverted to wildtype in vivo [146]. The second and third were a mother–child pair. The mother was infected via blood transfusion and the child via breastfeeding. Both displayed no loss of CD4^+^ T cells or any other symptoms for at least 13 years [147]. The final known infection by a Vpr mutant was via a needlestick containing ∆Vpr NL4-3. Over 10 years of close observation, the infected person’s viral load has usually been clinically undetectable (<20 copies per mL) and CD4^+^ T-cell count has been unaffected [148]. Combined, these results indicate that Vpr is a critical factor in HIV pathogenesis and transmission.

Because Vpr is packaged in the HIV virion [149], it has long been thought that it plays a role in early infection events. Packaging of Vpr is dependent on the p6 component of the Gag polyprotein, and Vpr is packaged at a nearly 1:1 molar ratio with Gag [150]. The conserved LXSLFG motif in p6 Gag is necessary and sufficient to package Vpr and the interaction between these proteins is direct [151]. Vpr localizes to the nucleus/nuclear envelope in PBMCs [152] and macrophages [153]. This observation led some to hypothesize that Vpr plays a role in viral integration, but recent studies have not observed a requirement for Vpr in first round infections of macrophages [14,142]. Early studies identified a host protein originally named Vpr binding protein or VprBP [154] that immunoprecipitates with Vpr, but the purpose of this interaction was unknown. This protein, which was later renamed DCAF1, was subsequently found to be required for nearly all of Vpr’s functions [155].

### 5.1. Vpr’s Role in Viral Replication

The first investigations of Vpr’s activity in infected cell cultures found that it did not have significant effects in CD4^+^ T cells [11,81]. Recently, there is evidence that Vpr enhances T-cell infection under certain conditions. Vpr causes ubiquitination and subsequent proteasomal degradation of helicase-like transcription factor (HLTF), a multi-domain, multi-functional protein that activates post-replication DNA repair [156,157]. Vpr-mediated degradation of HLTF enhances HIV replication in CD4^+^ T cells, but this effect is only apparent in competition assays [158]. Moreover, silencing HLTF did not fully restore replication by the Vpr mutant, indicating that Vpr has additional functions.

Vpr has a much stronger effect in macrophages, and this effect has been documented by numerous laboratories over decades [11,14,81,159,160,161]. Vpr enhances infection of human lymphoid tissue, which contains macrophages and T cells in a three-dimensional environment, but only when an M-tropic strain is used [81]. An early model proposed that Vpr enhanced nuclear import of the viral capsid and genome, but the evidence has been mixed. Assays using H9 cells demonstrated that Vpr enhanced the appearance of 2-LTR circles, a form of double stranded viral cDNA that only occurs in the nucleus [162]. Production of 2-LTR circles by Vpr-null HIV was rescued by addition of cytosol from HeLa cells, indicating that the nuclear import function of Vpr can be performed by unidentified host factors in some cell types. More recently, two studies have found that Vpr-null HIV does not display a defect in the first round of infection of monocyte-derived macrophages [14,142], indicating that Vpr is not required for nuclear import in these cells. This conclusion is also supported by an earlier finding that providing Vpr in trans, i.e., packaged in the virion but not encoded in the genome, does not fully rescue ∆Vpr virus; therefore, Vpr’s main functions occur after integration in macrophages [159].

### 5.2. Vpr and the Interferon Response

Through mechanisms that are incompletely understood, HIV induces a relatively weak interferon response, and Vpr contributes to this evasion. Vpr reduces transcription of IFNα [14], IFNβ, and *Mxa* by activating SLXcom, which leads to degradation of incompletely transcribed viral cDNA genomes [163]. It has also been demonstrated that Vpr acts via an unrelated pathway to specifically prevent induction of IFNβ [164]. In 293T cells, this is achieved by Vpr-mediated degradation of interferon regulatory factor 3 (IRF3), in which the HIV accessory protein Vif also plays a role [165]. Vpr has also been demonstrated to act further upstream by dysregulating TANK-binding kinase, which prevents activation of IRF3 [166]. Given that Vpr alters expression of numerous host proteins [167], it is possible that Vpr affects interferon signaling by multiple mechanisms.

### 5.3. Vpr-Mediated Cell Cycle Arrest

The most easily observable and most studied function of Vpr is arrest of the cell cycle at the transition from G2 to M phase [168]. This arrest at G2 phase increases activity of the HIV LTR and virion production [169], but it also induces apoptosis [170]. These opposing effects may be why Vpr does not have a clear positive or negative impact on infection of cycling CD4^+^ T cells in vitro [11].

Various models of Vpr cell cycle arrest have been proposed (Figure 5). There is broad agreement that the proximal cause of Vpr-mediated arrest is hyperphosphorylation of the host protein cdc2, a regulator of the DNA damage checkpoint, which prevents cdc2 from becoming activated [171]. Activity of cdc2 is controlled by at least two inputs, the phosphatase cdc25 and the kinase Wee1, and Vpr has been demonstrated to affect both. One group found that Vpr-mediated cell cycle arrest was abrogated in HeLa cells in which Wee1 had been silenced [172]. Another group later demonstrated that Vpr enhances Wee1 activity by directly binding to the kinase domain of Wee1 [173], although they found several Vpr mutants that bind and activate Wee1 but do not induce cell cycle arrest, indicating that binding Wee1 is not sufficient for cell cycle arrest.

There is significant evidence that Vpr also arrests the cell cycle via the other regulator of cdc2 activity, cdc25. During the normal cell cycle, tyrosine residues on cdc2 are de-phosphorylated by the phosphatase cdc25. Vpr has been shown to inhibit cdc25, leading to hyperphosphorylation of cdc2 [174]. Moreover, cdc25 is under the control of at least three upstream kinases: chk1, which is in turn controlled by ATM; chk2, which is controlled by ATR; and Srk1, which is activated by various stress responses [175]. Of these, only ATM is definitively not involved in Vpr-mediated arrest [174]. One study found that Srk1 is directly bound by Vpr, which increases Srk1-mediated phosphorylation of cdc25. This reduces cdc25 activity and prevents de-phosphorylation of cdc2 [176]. 

There is evidence from several studies indicating that Vpr can also act via ATR (ATM and Rad3 related protein), a host protein that arrests the cell cycle in response to DNA damage [177]. Inhibition of ATR by RNAi or overexpression of a dominant-negative mutant prevents Vpr-mediated G2 arrest [178]. The mechanism by which Vpr activates ATR has not been fully elucidated, but several studies have demonstrated potential pathways. One found that, in infected T cells, Vpr induces formation of replication protein A foci, which are known to activate ATR [179]. Another demonstrated that Vpr’s interaction with UNG2 promotes excision of uracil from viral cDNA [180]. Abasic sites left behind when UNG2 excises uracil have been demonstrated to activate ATR in cancerous cells [181]. It is possible that these are steps of the same pathway, i.e., that UNG2’s activity leads to the formation of replication protein A foci or vice versa, but this has not yet been tested.

A very recent study provided additional evidence that Vpr acts through the ATR pathway. This study demonstrated that the cellular protein CCDC137 is degraded by Vpr in cell lines and primary macrophages [182]. The authors found that silencing CCDC137 induces cell cycle arrest and boosts expression of viral proteins, two hallmarks of Vpr. Silencing CCDC137 induces formation of γH2AX foci [182], which is also known to be caused by activation of ATR [183]. How CCDC137 might activate ATR or whether it induces γH2AX foci by another means have not been determined. The mechanism by which CCDC137 enhances viral protein expression is also unknown and an important goal of future research

Although there is disagreement on how Vpr initiates the signal that leads to cell cycle arrest, it is universally agreed that it relies on DCAF1 to do so. The determinants within DCAF1 and Vpr necessary for Vpr to induce G2 arrest are well established [184,185,186,187]. Interestingly, one group identified mutations near the C terminus of Vpr that abrogate cell cycle arrest but do not affect DCAF1 binding, indicating that binding to DCAF1 is not sufficient to arrest the cell cycle [155]. 

Following a 2014 paper by Laguette at al., there has been considerable attention given to a model in which Vpr and DCAF1 act via components of the SLX complex (SLXcom), a multi-protein complex that resolves Holliday junctions following DNA repair by homologous recombination. SLX4 functions as a scaffold protein that recruits and assembles other subunits, including the endonucleases MUS81 and EME1. SLX4 expression was required for Vpr to induce G2 arrest in HeLa cells and MEFs [163]. This study also demonstrated that Vpr directly interacted with SLX4 and DCAF1, which led to decreased steady state levels of MUS81 and EME1 and activation of SLXcom. Finally, they demonstrated that this activation causes HIV reverse transcripts to co-immunoprecipitate with SLX4 and prevents induction of an interferon response. The authors proposed that evading IFN may be the true purpose of SLXcom activation by Vpr and that G2 arrest may be a side effect. This work was originally performed in cell lines, but a study in our laboratory confirmed that Vpr decreased steady state levels of MUS81 protein and *IFNA1* mRNA in primary macrophages and CD4^+^ T cells [14]. Mechanistic details of the pathway leading to cell cycle arrest have been debated by several groups. One study found that activation of SLX4 is not broadly conserved across isolates of HIV-1 and HIV-2 and that SLX4 was not required to mediate G2 arrest in U2OS and 293T cells [188]. Another group found that neither SLX4 nor DCAF1 binding was required for Vpr-mediated downregulation of MUS81 and EME1 and that downregulation of MUS81 and EME1 was not sufficient to arrest the cell cycle at G2 [189]. The differences between these findings and those of Laguette et al. may be due to differences in viral strains and cell lines used. Neither Fregoso and Emerman [188] nor Zhou et al. [189] investigated Vpr-mediated reduction of IFN, which may require SLX4. Follow-up studies focused on the mechanism of IFN evasion in primary cells would be highly informative.

### 5.4. Vpr-Mediated Degradation of Host Proteins

In addition to the host proteins implicated in Vpr-mediated cell cycle arrest described above, Vpr alters the expression of numerous other host proteins [167]. Several of these are direct targets, which Vpr ubiquitylates by using its cellular cofactor DCAF1. DCAF1’s normal function in the cell is to direct the activity of the DCAF1–DDB–Cullin4 E3 ligase complex, which ubiquitylates host proteins, usually to cause their degradation, but occasionally to regulate their activity [190]. Vpr simultaneously binds DCAF1 and its protein targets, changing the ligase complex’s substrate specificity.

It is well established that Vpr degrades UNG2 and SMUG1, two uracil deglycosylases [191,192], although the purpose of this is unclear. Degradation is mediated by Vpr binding directly to both UNG2 and DCAF1 [193,194]. Different studies of UNG2 have determined that its effect on viral replication can be negative, positive, or neutral. One group found that expression of UNG2 in producer cells had no discernable effect on infection of several types of targets cells, including multiple cell lines and monocyte-derived macrophages [195]. A study by another group found that UNG2 in target cells caused degradation of uracilated viral cDNA, indicating that UNG2 is a viral restriction factor, although this effect was only detectable when the dUTP:DTTP ratio and UNG2 activity in the target cell are both high [196]. Because expression of UNG2 is low in both primary CD4^+^ cells and macrophages [197], this restriction may not be active in the cell types most relevant to HIV pathogenesis. A third group found that, in addition to depleting UNG2 in infected cells, Vpr recruits UNG2 to virions, thus reducing the mutation rate of reverse transcription in target cells [198]. This study also demonstrated that a mutation in Vpr (W54R) that prevents binding to UNG2 increased mutation of HIV genomes in macrophages. Surprisingly, follow-up studies indicated that the deglycosylase activity of UNG2 is not required to reduce mutation of the HIV genome [199], and in fact UNG2 acts by forming a three-part complex between itself, Vpr, and replication protein A [197]. In sum, studies of the role of UNG2 in HIV replication indicate it is highly sensitive to conditions in both producer and target cells. Additional studies using primary cells as both producer and target, with UNG2 knockouts in each, would help resolve apparently contradictory findings in the literature.

In recent years, Vpr has been implicated in the regulation of TET2, a member of the TET family that regulates 5’ methylation of cytosine. In the absence of HIV and Vpr, DCAF1 monoubiquitylates TET2, which increases its affinity for chromatin [190] and therefore its activity. A follow-up study found that Vpr induces polyubiquitylation and subsequent degradation of TET2. Because TET2 inhibits transcription of the *IL6* gene, Vpr’s anti-TET2 activity increased IL-6 expression [200]. A later paper by the same group found that Vpr mediated degradation of TET2 prevented induction of interferon inducible trans-membrane protein 3 (IFITM3), which interferes with Env processing [142].

Very recently, additional cellular targets of Vpr that restrict infection have been identified. RNA-associated Early-Stage Antiviral Factor (REAF) was first identified as a factor that inhibits an early stage of infection by HIV-1, HIV-2, and SIV in cell lines [201]. A follow-up study found that silencing REAF in MDM enhanced infectivity by about three-fold and that Vpr prevented a spike in αREAF staining in the nuclei of MDM [202]. This result was demonstrated in MDM from two donors, in which the intensity and timing of the spike varied. Experiments in MDM from additional donors would help confirm the timing and mechanism of REAF’s action. 

### 5.5. Vpr Enhances Expression of HIV Env

Recently a series of papers by our group and others have reported that Vpr enhances steady state levels of HIV Env in monocyte-derived macrophages (MDM). The earliest report presented significant evidence that, in the absence of Vpr, all three forms of Env (gp160, gp120, and gp41) were degraded post-transcriptionally by an unidentified macrophage-specific host factor [14]. A pulse chase analysis demonstrated that Vpr had no effect on synthesis of Env, but Env protein was eliminated from the cell more quickly when Vpr was absent. This effect was inhibited by ammonium chloride but not MG132 indicating that Env degradation was lysosome-dependent. Importantly, we observed that Vpr stimulated virion release, but only when Env was expressed [14,88], suggesting that Vpr counteracts a restriction factor that directly targets Env.

A second study used microscopy to confirm that, in the absence of Vpr, Env and Env-containing virions were trafficked to lysosomes [88]. We also observed that Vpr-mediated enhancement of Env expression led to enhanced formation of virological synapses, which caused a substantial increase in the spread of HIV from infected MDM to co-cultured, autologous T cells [88]. Based on these observations and others [14,88], we hypothesized that Vpr counteracts a host factor that restricts Env expression, virion release, and viral spread by directly binding Env.

A report from another group confirmed that the loss of Vpr decreases steady state levels of gp160, gp120, and gp41 in HIV infected MDM [203]. They also observed this effect in monocyte-derived dendritic cells (MDDCs) and found that in MDDCs Env expression is rescued by treatment with bafilomycin (an inhibitor of lysosomes) and kifunesin (an inhibitor of ER-associated degradation (ERAD)). They concluded that Vpr prevents ERAD-mediated degradation of Env in MDDCs but did not directly test the mechanism of Vpr’s action in MDM. 

A study recently published by our group [89] identified MR as a restriction factor of Env that is counteracted by Vpr in MDM. We observed that Vpr decreased expression of MR protein by reducing levels of *MRC1* mRNA and confirmed an earlier report that the HIV accessory protein Nef also reduces MR surface expression [127]. Further we found that Env expression in infected MDM in the absence of Vpr was rescued by removing mannose-containing glycans on Env or silencing MR, conclusively demonstrating that MR is an anti-Env restriction factor. Additionally, silencing MR rescued HIV transmission from MDM to co-cultured autologous T cells in the absence of Vpr, indicating that, in our infection model, MR’s antagonism of Env restricts infection of both macrophages and T cells (Figure 6). To rule out contributions from known restriction factors, we assessed the expression of IFITM3, GBP5, and STING in HIV-infected MDM, with or without Vpr, and observed no consistent differences. Thus, we concluded that the degradation of Env observed in the absence of Vpr in our infection model was due primarily to the action of MR.

A third group recently reported that Vpr enhances Env expression in MDM [142], although their observations and conclusions differ from those reported by our group and Zhang et al. [203] in several key ways. Wang and Su [142] found that Vpr degrades the dioxygenase TET2, which prevents TET2 from demethylating the promoter controlling *IFITM3*, a known antiviral restriction factor. In the absence of Vpr, they observe a decrease in the steady state levels of the cleaved forms of Env (gp120 and gp41) but not the unprocessed form (gp160). The effects on gp120 and gp41 were eliminated when IFITM3 was silenced and the authors propose that IFITM3 interferes with cleavage of Env. The observation that the steady state level of gp160 is unaffected is in contrast to findings from our group and others [203] that gp160 is destabilized in the absence of Vpr, although to a lesser degree than gp120 and gp41 [14,89]. Moreover, Wang and Su report that Vpr did not enhance virion production in infected MDM, which is in contrast to our observations [14,88,89]. 

These differences may indicate that our groups are observing effects of two different restriction factors, which may be due to differences in gene expression in our macrophage models. Wang and Su observed that expression of IFITM3 was constitutive, whereas we observed IFITM3 expression only after HIV infection [89]. Importantly, we observed inconsistent effects by Vpr on IFITM3 expression; in four side-by-side infections with or without Vpr, we found that Vpr reduced IFITM3 expression is only one case [89]. In two cases, Vpr was associated with greater IFITM3 expression, likely due to increased infection in the culture. These substantially different observations suggest that the mechanism by which IFITM3 expression is regulated differs significantly between our MDM model and that used by Wang and Su. To minimize activation of macrophages, our group uses only low endotoxin FBS, which may explain why IFITM3 is not detected in our uninfected MDM. There are several other differences in our stimulation protocol, such as types and concentrations of stimulating cytokines, cell density, and timing of media changes, that may cause differences in expression of IFITM3, MR, and proteins that regulate their expression. Indeed, others have reported that different stimulation conditions can alter expression of numerous proteins in MDM [53].

It is difficult to conclusively determine which MDM stimulation conditions produce the best model of in vivo macrophages, partially because many forms of macrophages are infected by HIV. Therefore, it is possible that both MR and IFITM3 can restrict HIV Env in vivo under different conditions and that HIV Vpr counteracts both of these factors. There is significant evidence from other groups that HIV antagonizes MR in vivo and that this effect is physiologically important. It has previously been observed that HIV infection decreases phagocytic function in alveolar macrophages [125], which correlates with reduced MR expression [72]. Consistent with this finding, there is reduced expression of MR on glial cells in the brains of SIV infected macaques [126]. Given these observations, we believe that HIV infection of macrophages is an important aspect of HIV pathogenesis. Defining the complex interactions between Env, Vpr, and macrophage-specific restriction factors such as MR is an important objective of future research.

## Figures and Tables

**Figure 1 viruses-12-00809-f001:**
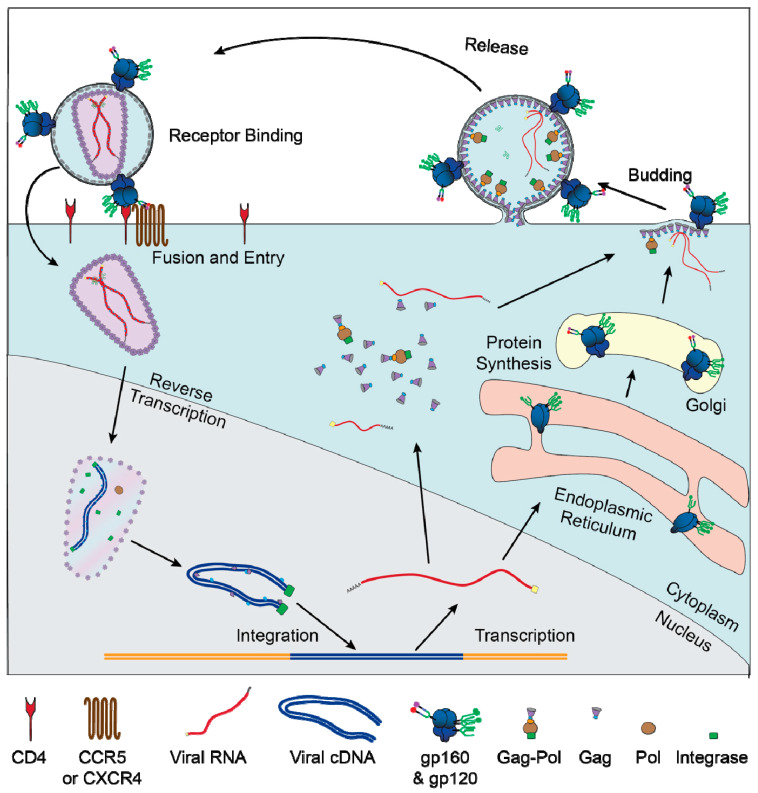
Replication cycle of HIV: graphical depiction of the major events of the HIV replication cycle. Viral entry is mediated by binding of Env to CD4 and a co-receptor (CXCR4 or CCR5). After the viral membrane fuses with the plasma membrane the capsid and RNA genome are transported into the nucleus. The viral genome is reversed transcribed by Pol and integrates into the host genome. The host RNA polymerase transcribes RNA copies of the genome, which are exported in an unspliced form to produce Gag and Gag–Pol or various spliced forms to produce the other viral proteins. Unspliced RNA genomes are also packaged into newly formed virions. Most viral protein translation occurs in the cytoplasm, but Env gp160 is translated into the lumen of the rough ER and transported through the secretory pathway, where it is glycosylated and cleaved by furin into gp120 and gp41.

**Figure 2 viruses-12-00809-f002:**
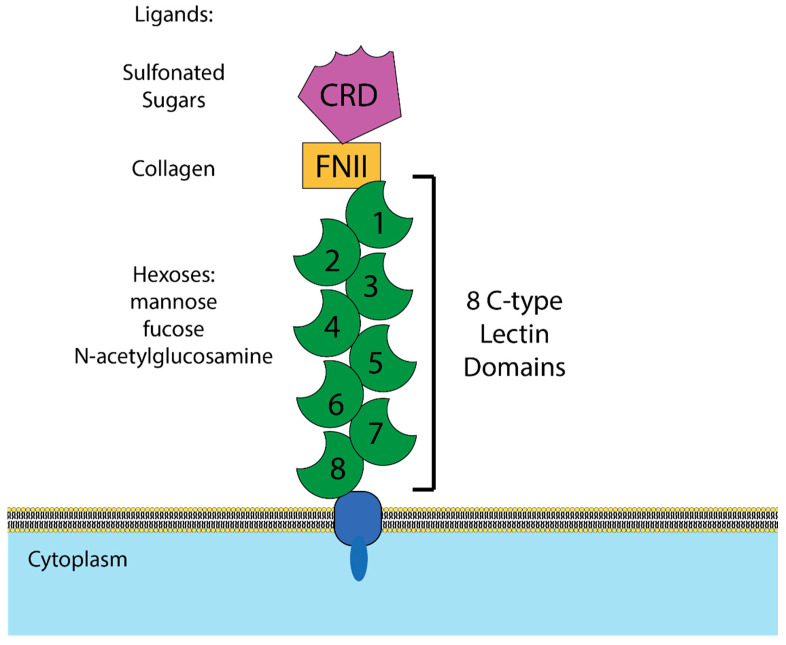
Structure of mannose receptor (MR). Graphical depiction of the domains of MR and the ligands bound by each domain. CRD = cysteine rich domain, which binds sulfonated sugars. FNII = fibronectin type II domain, which binds collagen. C-type lectin domains = calcium-dependent domains that have been demonstrated to bind mannose, fucose, and N-acetylglucosamine.

**Figure 3 viruses-12-00809-f003:**
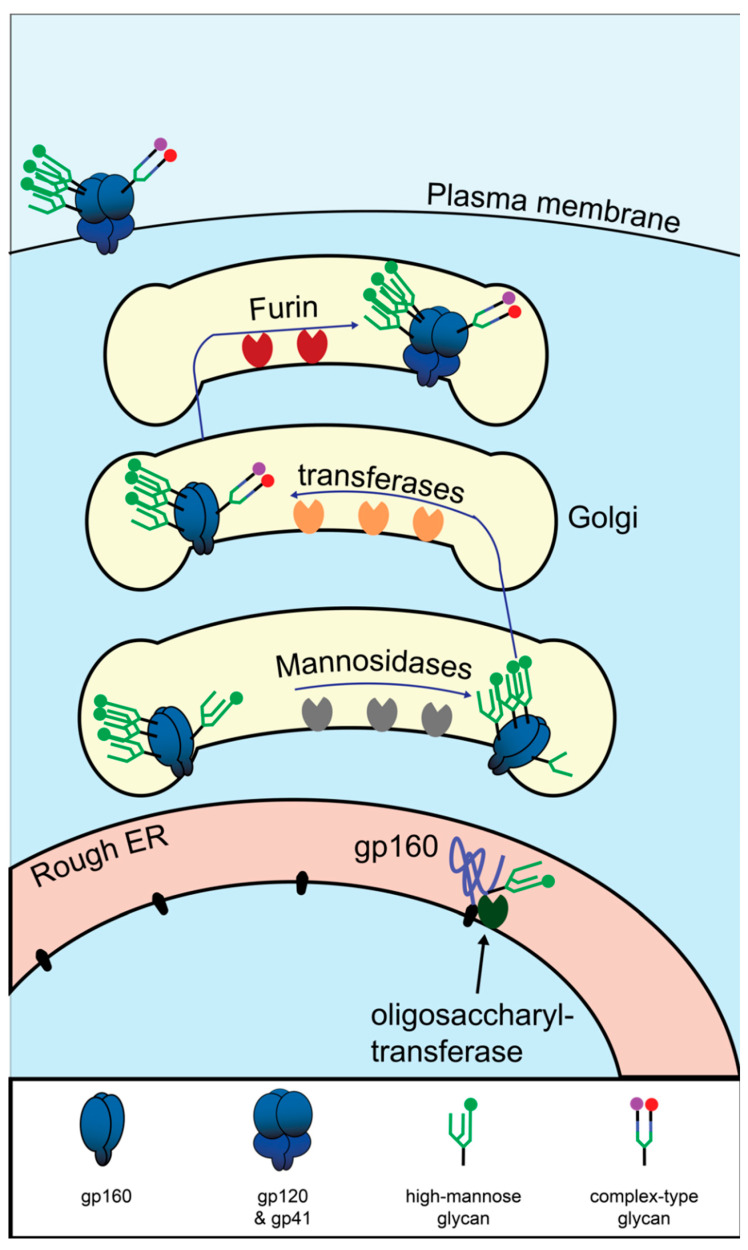
Biosynthesis HIV Env in the secretory pathway. Graphical depiction of translation and glycosylation of Env in the endoplasmic reticulum, followed by post-translational modifications in the Golgi. Mannose oligomers are depicted in green. The mannose patch is depicted as three densely packed glycans. For a detailed depiction of the monomers composing the glycans, see Figure 4.

**Figure 4 viruses-12-00809-f004:**
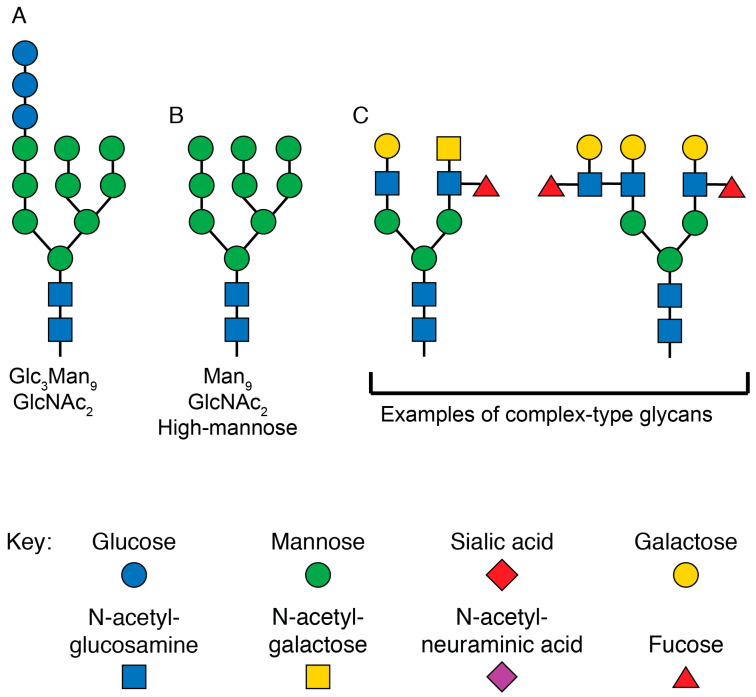
Structures of glycans at various stages of N-linked glycosylation. (**A**) Graphical depiction of the glycan structure transferred to all N-linked glycosylation sites by the enzyme complex oligosaccharyltransferase in the rough ER. The three terminal glucose monomers are removed before exiting the ER. (**B**) Graphical representation of the high-mannose glycan at N-linked glycosylation sites as newly synthesized proteins enter the Golgi. For certain sites on HIV Env, this is the final form of the glycan. (**C**) Graphical representation of two complex-type glycans that are present on mature eukaryotic proteins. These are just two of the dozens of forms the final mature glycan structure can take. The sugar monomers are drawn according to the updated recommendations for symbol nomenclature for glycans [133].

**Figure 5 viruses-12-00809-f005:**
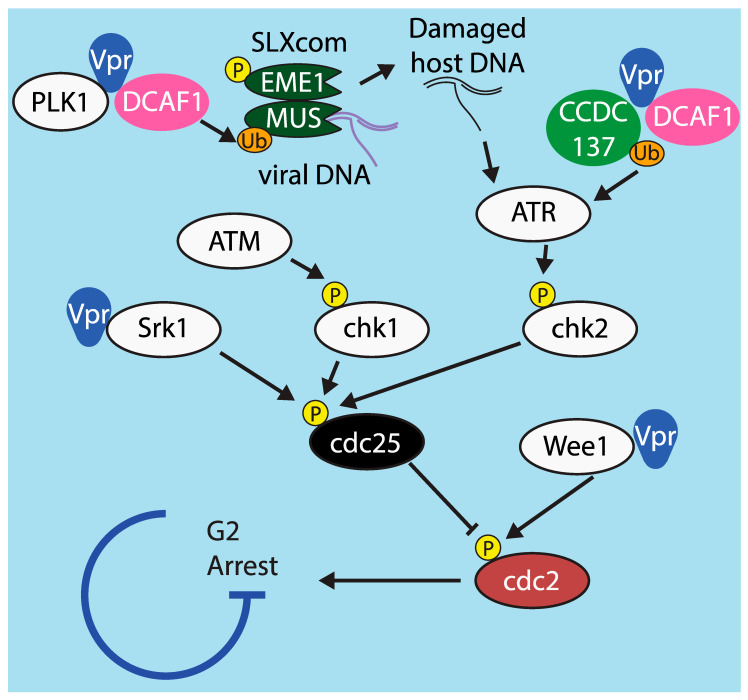
Model of G2/M cell cycle arrest and mechanisms by which it is affected by Vpr. Graphical depiction of the cdc2-mediated cell cycle arrest pathway, including points at which Vpr has been demonstrated to alter signaling in order to promote arrest.

**Figure 6 viruses-12-00809-f006:**
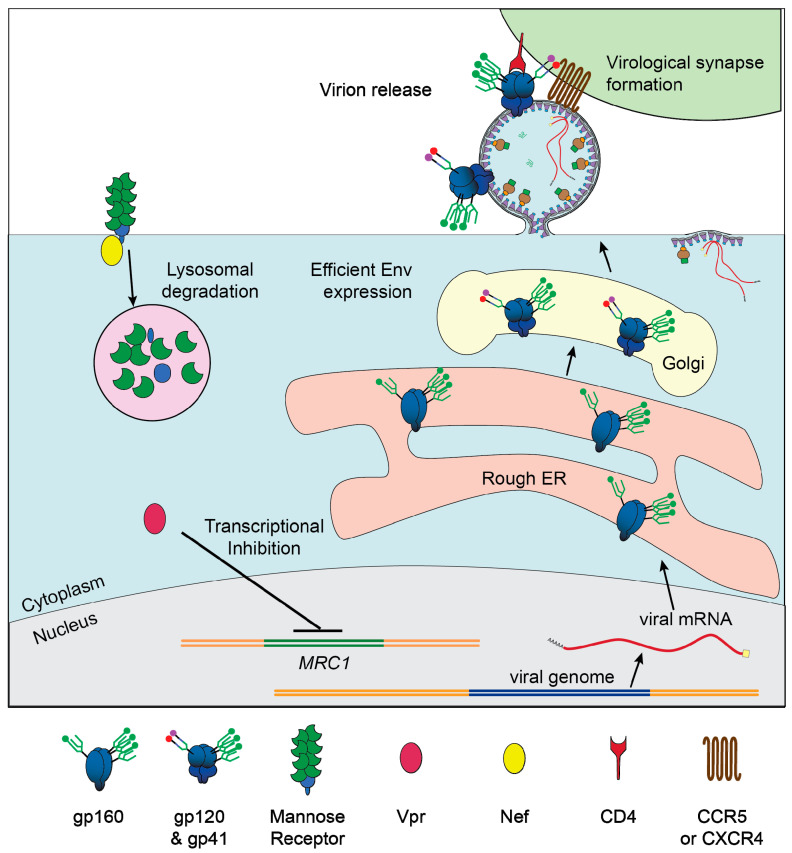
Vpr- and Nef-mediated downmodulation of MR lifts restriction of HIV in macrophages. Graphical depiction of our proposed model in which Vpr reduces transcription of *MRC1*, the gene that produces MR, and Nef binds to MR, to remove it from the cell surface. Low MR expression allows for high Env expression, efficient virion release, and formation of virological synapses between infected macrophages and uninfected CD4^+^ T cells.

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
