# Peer review of "Vpr Is a VIP: HIV Vpr and Infected Macrophages Promote Viral Pathogenesis"

_viruses, 2020, doi:10.3390/v12080809_

Round 1
Reviewer 1 Report
The first goal of this manuscript by Lubow and Collins is to review the general role played by macrophages as target cells of HIV-1 for different aspect of viral pathogenesis. In the second part of the manuscript, the authors provide a good update of the literature on recent developments of the specific role of the viral Vpr auxiliary protein for efficient replication in macrophage targets, and then discuss in more details recent findings they obtained regarding functional interactions between Vpr, the viral Envelope and the Mannose receptor for a better understanding of the contribution of Vpr and macrophage infection to HIV-1 pathogenesis.
The manuscript is rather well written and usually provide exhaustive information and details about the specific role and functions of Vpr for efficient HIV-1 replication in macrophages. While better transitions between the different parts and chapters of the manuscript would facilitate reading of the manuscript, a few modifications and additional references should improve the intrinsic quality of this literature review.
Specific points:
1) In addition to the Mannose receptor, the authors should also mention other C-type lectin receptors, i.e. DC-SIGN and Langerin, as well as the sialic acid lectin receptor SIGLEC-1 (CD169), which have been also involved in HIV-1 capture by target myeloid cells.
2) Page 5, lanes 175-176, the authors could mention that other cells of the myeloid lineage including dendritic cells but also osteoclasts, are also targeted by HIV-1 in addition to macrophages.
3) Pages 5 and 6, lanes 207-216, the authors wrongly referred to "transgenic" mice to describe the results recently published by Honeycutt et al (Refs 93 and 94 of the present manuscript) to show that myeloid cells are sufficient to sustain HIV-1 replication in vivo independently of CD4+ T cells. However, in these publications, Honeycutt and coll. did not use transgenic mice but an original "Humanized" myeloid-only mouse model.
4) Page 7, lanes 228-234, the authors could mention works and give additional references (Mazzolini, 2010; Verollet, 2010; Dumas, 2015; Verollet, 2015) regarding how Vpr has been involved in alterations of specific macrophage functions, such as phagocytosis, motility and migration, when infected by HIV-1.
5) Page 7, lanes 244-257, some works and references from the respective groups of Quentin Sattentau (Baxter, 2014) and Serge Benichou (Bracq, 2017; Xie, 2019) are missing, regarding alternative mechanism for HIV-1 cell-to-cell spreading in macrophages.
6) Page 13, lane 473, the authors should also mentionsubsequent studies, and not only one, by Guenzel et al. (2012), and Herate et al. (2016) regarding involvement of interaction of Vpr with UNG2 for efficient reverse transcription and HIV-1 replication in macrophages.
Reviewer 2 Report
General impression
Aims are set out clearly in the abstract: the authors intend to “review the role of macrophages in HIV infection and describe complex interactions between viral proteins and host defences”, centred on the action of Vpr. Good background information is provided prior to starting into details of the study. For the most part conclusions agree with the data used to support them and biases originating from technical aspects of studies (i.e. the choice of models) are highlighted. The authors also make some suggestions for future directions of study however these are limited. While the recent findings about interactions between mannose receptor, Vpr and Env are interesting and have clear implications for HIV infection and pathogenesis, I feel this is presented as being the main axis through which Vpr carries out its role which is not certain at this point and overlooks many other targets of Vpr. Although the writing is clear and concise, the layout is not always ideal.
Major layout suggestions:
Macrophage introduction is lengthy and not all the information presented seems to be relevant to the scope of the review which was set out in the abstract. The authors might consider starting the review by an overview of HIV which would help set up a more focused section on macrophages. The information contained in section 1 should be summarized (particularly sections 1.1. and 1.4.) and sections 1.2. and 1.3. focused on macrophages in HIV infection (using information contained in sections 3.2 and 3.3. for example). This could be followed by the important background material on mannose receptors and Env before looking into their interactions with Vpr. I would also suggest that sections 5.2. and 5.3. be placed before 5.1. to introduce Vpr, its importance in infection and its mechanism of action before exploring Vpr associated phenotypes. Describing the importance of Vpr in infection with the understanding that it primarily affects macrophages also further supports the earlier points about the relevance of macrophages in HIV infection.
Section by section
1. Macrophages in health and disease
1.1.
While interesting and well written, the information is not that linked to the topic at hand. Good to point out the difference between monocyte derived and tissue resident macrophages as well as their origin. The details about macrophages in each tissue seem superfluous as they are not mentioned in any later writing they could just be condensed to the first sentence: “There are numerous types of macrophages in the human body that are specialized for their particular role and anatomical location.” (line 28).
1.2.
Good information on the role of macrophages in phagocytosis and cross-presentation. Perhaps it would be appropriate to at least mention the role of macrophages in inflammatory responses, as well as macrophage polarization (e.g. reviewed in 10.3390/ijms19061801) which is central to many pathologies. It seems important to describe their role more broadly in the context of the immune system. It could also be interesting to mention that phagocytosis is impaired in HIV infected macrophages, which could have implications for secondary infections with other pathogens (https://doi.org/10.1038/mi.2013.127, PMID 7751654, https://doi.org/10.1182/blood-2009-12-259473, https://doi.org/10.1038/ncomms7211).
1.3.
Interesting to place macrophages in the context of infection. I would however use this section to address the evidence of HIV infection of macrophages which is only discussed in 3.2. w
1.4.
It is important to define the models used in the studies presented. However, the arguments being made here about variability are further repeated in lines 536 to 544. Could the models not be described in text with the studies that use them?
2. Mannose receptor
2.1.
Given the relatively small role of mannose receptor in HIV infection there is way too much detail here
2.2.
Again as of comment above there is too much detail here on mannose receptor and interaction with other pathogens. It is better to stick to HIV or SIV considering the remit of the review article. Lines 159 – 166. The authors might want to comment on apparent conflicting evidence. Mannose receptor being implicated in increased HIV infection would suggest it has a positive role in infection, so why would it be targeted by Vpr? Could be interesting to search literature further for how phagocytosis affects HIV infection of macrophages. One study suggests phagocytosis of infected and dying CD4 cells by macrophages is a route of infection of macrophages (https://doi.org/10.1186/1742-4690-8-S2-O31)
3. Human Immunodeficiency Virus
3.1.
Good figure for HIV life. However there is controversy on whether reverse transcription occurs in the cytoplasm. Indeed there is now strong evidence to suggest that it occurs in the nucleus. This will be an important point to clarify if the review is to be timely. Glycosylation (the blue and green on gp120) detracts from the message. There is no maturation event shown. Could be helpful to include a figure for HIV genome architecture. Lines 195-196 stating that accessory proteins “do not actively participate in replication processes but […] alter the host cell environment to maximize virion production and persistence” seems slightly reductive. The definition of accessory proteins as “non-essential” is based on in vitro models, but evidence suggests they play an important role in in vivo infection and pathogenesis – this is alluded to in section 5 where it is stated that Vpr mutants are extremely rare and are selected against.
3.2.
Lines 204-205 mention theories that macrophage infection is of limited importance in natural infection. I think the best rebuttal to this is the role of macrophages in HIV pathogenesis, particularly in secondary infection, which only comes at the next paragraph. It is also noteworthy that viral load is associated with disease progression in the case of AIDS/HIV which is not necessarily made clear here. Line 254 last word “in” should be “is”. The authors could comment further on the relative importance of virological synapses versus cell free virus. While there is significant evidence for it, it is not entirely established that cell-cell transmission is the main mode of transmission of HIV.
Please reference line 244/245. What is the significant evidence in vivo. Why is it particularly important in macrophages? Not T-cells in the lymph node. Is vpr important in cell-to cell transmission? Does it overcome the need for vpr? Why the need for so much detail in this review?
Lines 268-270. Perhaps it would be better to cite evidence that these macrophages are actually infected? There are plenty of studies in vitro some of which are reviewed in this chapter but it would be helpful to include a more in depth review of this data.
4. Env biosynthesis
I think it’s a bit late here to introduce the env protein in this detail. Nor is the detail required for a review that is supposedly focussed on the role of vpr in macrophage infection. Env has already been covered sufficiently with the life cycle. In general there is way too much coverage on env biosynthesis. That is not to say it is not interesting and well written but that it is not helpful in a review that addresses Vpr and macrophage infection.
Line 293 could use a reference to a review on Env biosynthesis (e.g. 10.1016/j.jmb.2011.04.042). This would allow the authors to shorten this section, and redirect readers to further information about it should they require it.
5. The HIV accessory protein Vpr
Introductory paragraph mentions mostly evolutionary evidence of Vpr’s crucial role in infection. That evidence is strong and useful but maybe it could be supplemented by ex vivo evidence such as reduced titres in Vpr mutant viruses. Could also cite Vpr requirement for efficient replication in macrophages (re-use reference 102 here, also reviewed in https://doi.org/10.1016/j.chom.2008.04.008). (Could use information from section 5.2.).
Line 345. Was the reversion to wt vpr co-incident with disease progression? It may be useful here to distinguish between being crucial for “infection” or pathogenesis, viral load and disease progression.
Line 445 to 448 is a bit of a jump in conclusion. There are many papers that would contradict this conclusion. Here it might be more interesting to discuss how much vpr is actually packaged by p6 into the virion. There are a number of estimates in the literature.
5.1.
Good information in section on cell cycle arrest, although I find that the importance of vpr-DCAF1 in this phenotype comes late in the section, making its importance less obvious to the reader. Perhaps it would be useful to describe this interaction between Vpr and DCAF1 before talking about Vpr induced phenotypes.
Further to previous points the role of Vpr in viral replication should be part of the introductory paragraph.
5.3 , 5.4.
Again, I believe it would be logical to place the vpr-DCAF1 mechanism earlier in the section (just after introductory paragraph) to help describe further phenotypes such as cell cycle arrest and proteome remodelling (including IFN). It is unclear why so much focus has been drawn to the targeting of UNG2 and SMUG1. Their role in infection is far from being clear. Perhaps this section could be used to look at other Vpr targets with better known antiviral properties (e.g. REAF/RPRD2). I think the point made in line 458 should be made more prominent: Vpr has a broad role in remodelling the protein landscape of the cell. Restricting its role to a select few targets seems to reduce the importance of systemic changes that Vpr could be inducing.
5.5.
Lines 494-495 are unclear to me, what is meant by the “Vpr-sensitive factor”? Line 501 I don’t understand where in the references provided is the evidence that shows that the Vpr target binds Env directly. The effects of Vpr and Nef are broad and could easily be indirectly affecting MR expression or function. Nef’s effect on MR is also well documented, but the authors present Vpr as the main actor of this counter-restriction. Lines 515-516 “figure 6” refers to the wrong figure in the reference, it should be figure 7. The claim that “silencing MR rescued HIV transmission from MDM to co-cultured, autologous T cells in the absence of Vpr, indicating that MR’s antagonism of Env restricts infection of both macrophages and T cells” should be tempered by saying this is true in their particular model – which perhaps needs to be described. Lines 518-519 is a really strong claim that degradation of Env in absence of Vpr is solely due to MR action. The authors have only tested a handful of restriction factors, specifically ones that target egress from the cell. Therefore this statement is not entirely supported by evidence considering the broad role of Vpr, which may result in indirect effects on Env biosynthesis. The way they have “ruled out” the action of GBP5 and IFITM3 is not very convincing, especially considering other groups have found a role for IFITM3, which they talk about later.
The conclusion which focuses on MR for future research is rather limited. There have been at least three papers describing Vpr mediated restrictions which have not been discussed which should be included (Zhang and Bieniasz, 2020, Gibbons et al 2020 and Kang et al 2020).
